# Inertial Data-Based AI Approaches for ADL and Fall Recognition

**DOI:** 10.3390/s22114028

**Published:** 2022-05-26

**Authors:** Luís M. Martins, Nuno Ferrete Ribeiro, Filipa Soares, Cristina P. Santos

**Affiliations:** 1Center for MicroElectroMechanical Systems (CMEMS), University of Minho, 4800-058 Guimarães, Portugal; lmartins2116@gmail.com (L.M.M.); a86677@alunos.uminho.pt (F.S.); cristina@dei.uminho.pt (C.P.S.); 2LABBELS—Associate Laboratory, 4710-057 Braga, Portugal; 3LABBELS—Associate Laboratory, 4710-058 Guimarães, Portugal; 4MIT Portugal Program, School of Engineering, University of Minho, 4800-058 Guimarães, Portugal

**Keywords:** activity recognition, falls, feature selection, dataset fusion, Machine Learning, deep learning

## Abstract

The recognition of Activities of Daily Living (ADL) has been a widely debated topic, with applications in a vast range of fields. ADL recognition can be accomplished by processing data from wearable sensors, specially located at the lower trunk, which appears to be a suitable option in uncontrolled environments. Several authors have addressed ADL recognition using Artificial Intelligence (AI)-based algorithms, obtaining encouraging results. However, the number of ADL recognized by these algorithms is still limited, rarely focusing on transitional activities, and without addressing falls. Furthermore, the small amount of data used and the lack of information regarding validation processes are other drawbacks found in the literature. To overcome these drawbacks, a total of nine public and private datasets were merged in order to gather a large amount of data to improve the robustness of several ADL recognition algorithms. Furthermore, an AI-based framework was developed in this manuscript to perform a comparative analysis of several ADL Machine Learning (ML)-based classifiers. Feature selection algorithms were used to extract only the relevant features from the dataset’s lower trunk inertial data. For the recognition of 20 different ADL and falls, results have shown that the best performance was obtained with the K-NN classifier with the first 85 features ranked by Relief-F (98.22% accuracy). However, Ensemble Learning classifier with the first 65 features ranked by Principal Component Analysis (PCA) presented 96.53% overall accuracy while maintaining a lower classification time per window (0.039 ms), showing a higher potential for its usage in real-time scenarios in the future. Deep Learning algorithms were also tested. Despite its outcomes not being as good as in the prior procedure, their potential was also demonstrated (overall accuracy of 92.55% for Bidirectional Long Short-Term Memory (LSTM) Neural Network), indicating that they could be a valid option in the future.

## 1. Introduction

The recognition of Activities of Daily Living (ADL) has been a widely debated topic of study for the past several years with applications in a vast range of fields, from medicine to supervision of a persons’ driving style, and even to sports training analysis, passing through surveillance [1,2]. Several activities are being recognized nowadays. ADL related to human locomotion, such as walking, running, moving up and down stairs, or just sitting or lying down, are identified in several papers [2,3,4,5,6]. Other activities involving finer gestures with the upper limbs, such as driving, talking on the phone, or eating are also addressed [3,7]. Fall detection [8], recognition of sedentary behavior [9] and comfort in smart homes [10] are just a few more examples of automatic ADL recognition applications. The richness and diversity of human activities, as well as the large dimensionality of the data gathered, make the recognition of human activities a difficult but promising endeavor [11]. Furthermore, intraclass variability (the same activity may differ from one subject to the other) and interclass similarity (various activities may exhibit similar forms) make recognizing human activities even more challenging [12].

The identification of human activities can be accomplished by processing signals or image sequences gathered from one or more sensors, such as examining videos and photos from cameras or motion data from inertial sensors [11]. When compared to video and environmental sensor-based systems [13], wearable sensors show to be a suitable option for ADL recognition in uncontrolled environments due to their small size, low cost, and adaptability [5,6]. Several studies on activity recognition used wearable sensing technologies [6], with simpler wearable systems relying on a single accelerometer sensor, and others combining different sets of sensors on different parts of the body to fully capture the subject’s movements [5,14]. Moreover, researchers have been creating various publicly available benchmark datasets to evaluate human activity recognition algorithms [5]. The UCI HAR [15], Sisfall [16], Opportunity [17], and PAMAP2 [18] are some of the most commonly used public datasets to train and evaluate inertial sensor-based ADL recognition algorithms.

Offline activity recognition is the most basic method of activity recognition [14]. Sensors are placed in the surroundings or directly on the subject in most applications. Data are gathered, labeled, and supplied to classification algorithms for activity recognition [5,6,14]. Most offline approaches rely on supervised Machine Learning (ML) models for activity recognition, such as Support Vector Machine (SVM), Decision Trees (DTs) and K-Nearest Neighbors (K-NN) [6,14]. Moufawad et al. [19] used several features in a DTs algorithm to differentiate nine classes of ADL, and an overall accuracy of 97% was obtained across all activities. An accuracy of 80% for a set of 14 ADL classified was attained by Gomaa et al. [4], and the smallest average sensitivity and specificity achieved by the proposed Random Forest (RF) classification algorithm were 81% and 98%, respectively. Two ML algorithms were implemented by Gupta et al. [20] for the classification of six classes of ADL—Naive Bayes (NB) and K-NN. The overall accuracy of the system was about 98% from both the classifiers and showed accuracy of more than 95% for all the activities. Awais et al. [21] used an SVM classifier with RBF kernel to classify the four activity classes with data acquired from several wearable sensors. Among all single-sensor solutions, the best performance was accomplished by the sensor at the lower back (L5), with an F-measure of above 80%, and there was a notable improvement in the performance of 7.3% from single-sensor solution to a two-sensor solution.

Moreover, on several benchmark datasets used by ML-based solutions, conventional (CNN and LSTM) and hybrid (CNN-LSTM) Deep Learning (DL) models have lately shown improved and promising performances, cutting down the time spent on data processing and feature extraction [5,14]. Experimental results attained by Wang et al. [8] show that lightweight neural networks have obtained better results than ML methods previously used in fall detection. The best result was attained with the supervised Convolutional Neural Network (CNN), which achieved an accuracy beyond 99%. Despite the great Fall Detection accuracy results, other activities were not recognised. Altuve et al. [2] used sliding window-based segments and several sequential Long Short-Term Memory (LSTM) neural network to identify six human activities, obtaining 92.91% average accuracy. A Many-to-One LSTM network architecture was used by Chung et al. [22] for the recognition of nine ADL and the results show that activity gathered from inertial data with a sampling rate as low as 10 Hz from four sensors is sufficient to recognize ADL with 93% accuracy. In the work developed by Murad et al. [3], unidirectional, bidirectional, and cascaded architectures based on LSTMs were created, and their effectiveness was evaluated on several benchmark datasets. As it can be observed, overall accuracies over 92% were attained for each dataset, which proved the efficiency of the created models for a broad range of activity recognition tasks. A three-part modular system based on CNNs was proposed by Gil-Martín et al. [23] in order to classify 12 different physical activities. Results show that when applying post-processing techniques, isolated classification errors were reduced, improving the accuracy from 89.83% to 96.62%. Table 1 describes the datasets used by the analyzed works as well as the results obtained by their Deep Learning algorithms.

For the most part, the results achieved by Artificial Intelligence (AI)-based algorithms for ADL recognition are highly satisfactory. However, the number of activities recognized is still generally small, with the analyzed state-of-the-art detecting a maximum of 14 ADL (in this case, with 80% accuracy) [4]. Moreover, the amount of data used, whether through data acquisition protocols or through public datasets use, is small, since the largest public dataset identified in our research contains data from 38 people [16], and the average number of subjects found is less than 22, and with low variability. As a result, there is a greater demand for large-scale data collection campaigns. Studies in the field of merging datasets and pre-processing pipelines are also required in order to properly combine and decrease disparities between data obtained from various sources [27]. Another issue discovered is that the validation processes used in the algorithms are not stated or are inadequately explained in some research. This may cause the developed models to perform well just on the used dataset, but they perform poorly when evaluated on new datasets [28]. Furthermore, despite the fact that some articles implement feature selection techniques, they do not indicate which features and information are most essential for ADL recognition.

This manuscript addresses the aforementioned drawbacks and, in addition, addresses a higher number of ADL than any other of the works analyzed before, recognizing a total of 20 classes, including transitional activities (Sit-to-Stand, Stand-to-Sit, Lying-to-Stand, Stand-to-Lying, pick objects from the ground, bending and turning) and falls (16 ADL and four types of falls). Furthermore, this manuscript aims to answer the following research questions concerning recognition/classification of 16 ADL and four types of fall events from data collected from waist-located inertial sensors: (i) Which is the most suitable classifier and what are the most relevant features? and (ii) Which approach, ML or DL-based models, presents better performance?

With these objectives in mind, an AI-based framework was developed that allows for the benchmarking of several classifiers, including Discriminant Analysis (DA), K-NN, Ensemble Learning, DTs, SVM, CNN, mono and bidirectional LSTM, and CNN-LSTM neural networks. Simultaneously, feature selection algorithms are used to extract only the relevant features from the dataset’s lower trunk inertial data. Furthermore, a dataset fusion and normalization procedure is applied in order to gather a large amount of data to improve the robustness of the suggested algorithms during the several validation steps.

The remainder of this paper is organized as follows. Section 2 explains the inertial data-based dataset fusion and normalization process, feature extraction and selection processes, and finally, the comparative analysis carried out during the models building and evaluation. Then, Section 3 presents multiple results obtained with the proposed methods for validation, evaluation and best model choosing, followed by comparative studies, which try to uncover how changes in window size and model type can influence the final performance of classification models. Finally, Section 5 concludes the document, offering some remarks of the work developed along with directions for future improvements.

## 2. Methods

### 2.1. Public Dataset Fusion and Normalization

The proposed algorithm makes use of a large number of publicly available datasets to validate activity recognition models based on inertial data collected from the participants’ waists. Thus, a vast dataset was built from the fusion and normalization of several ADL public datasets. This process, to the best of our knowledge, has not been carried out before in any other works of the kind, despite its importance being highlighted multiple times [27,28]. Datasets that met certain requirements were searched: (i) be publicly available online for download; (ii) contain inertial data collected from the lower trunk (back) of at least accelerometers and gyroscopes; and (iii) contain postural daily activities and/or fall events. The research carried out resulted in the gathering of the public datasets and the three other team-own datasets, which are described in Table 2. The gathered public datasets were as follows.

Sisfall [16]: Data acquired with 23 healthy young adults (19–30 years, 149–183 cm, 42–81 kg) and 15 healthy elderly participants (60–75 years, 150–171 cm, 50–102 kg) with a device composed of two types of accelerometer and one gyroscope fixed to the waist of the participants, who performed 19 ADL and 15 fall types.FallAllD [29]: Data acquired from 15 healthy subjects (21–53 years, 158–187 cm, 48–85 kg) who used 3 devices equipped with an accelerometer, a gyroscope, a magnetometer and a barometer. A total of 44 classes of ADL and 35 classes of falls were performed.FARSEEING [30]: Large-scale collaborative database to collect and share sensor signals from real-world falls. Real fall data are acquired from either 2 locations: waist or thigh, and the acquisition devices are equipped with up to 3 sensors, namely accelerometer, gyroscope and magnetometer.UCI HAR [15]: Dataset recorded from 30 healthy subjects (19–48 years) by using a waist-mounted smartphone with an embedded 3-axis accelerometer, gyroscope, and magnetometer. This dataset contains six classes of ADL: walking, ascending stairs, descending stairs, sitting, standing, and laying.Cotechini et al. [31]: Dataset acquired from 8 healthy subjects (22–29 years old, 173–187 cm, 60–94 kg) using a wearable device containing a 3-axis accelerometer and gyroscope, tied to the subject’s waist, that recorded subject’s acceleration and orientation. Subjects simulated 13 typologies of falls and 5 types of ADL.UMAFall [32]: A dataset acquired from a total of 17 healthy subjects (18–55 years, 50–93 kg, 155–195 cm). Accelerometer, gyroscope and magnetometer data were colected from five wearable sensing devices, located on the subject’s chest, waist, wrist, ankle and pocket. The participants performed 8 different ADL and 3 different typologies of falls (except by those older than 50 years, who did not perform falls).

The used team-owned datasets were as follows.

+Sense [33]: Dataset with data acquired from 10 healthy subjects (44.02 ± 16.42 years, 67.5 ± 16.06 kg, 172 ± 7.93 cm) and 40 subjects with Parkinson’s disease (64.00 ± 10.60 years, 69.93 ± 11.41 kg, 165.93 ± 8.65 cm). A waist-mounted waistband, equipped with an accelerometer, a gyroscope and a magnetometer recorded subject’s data in walking activity protocols.SafeWalk [34]: Dataset acquired with 12 healthy subjects (25.33 ± 6.33 years old, 66.92 ± 10.07 kg, 1.74 ± 0.11 m). Five IMUs were attached to the lower back, both back thighs, and to both feet of the subjects, who performed walking trials and front fall events.InertialLab [35]: Dataset which includes data from 11 able-bodied subjects (24.53 ± 2.09 years old, 171 ± 10 cm, 65.29 ± 9.02 kg). Gyroscopes and accelerometers were attached to six lower limbs and trunk segments. Walking in varying speed and terrain (flat, ramp, and stairs) and including turns were the activities carried out by the subjects.

The fusion of these various datasets, which include information from both adult and elderly subjects, as well as healthy and unhealthy individuals, served two purposes. First, to create a large and diverse dataset suitable for training ML models. The second goal was to develop AI-based models capable of recognizing ADL regardless of the subject’s age and health condition.

A global dataset containing 6702 files covering a total of 20 ADL and falls conducted by 180 subjects (age = 33.60 ± 16.84 years, weight = 69.98 ± 10.99 kg, height = 168.99 ± 9.42 cm) was generated. This vast amount of data is crucial for validation purposes since it is far larger than any other dataset used in the AI-based ADL recognition research carried out. Furthermore, the global dataset presents a balanced distribution regarding the subject’s gender (M = 54%, F = 46%), also containing data from both young adults and elderly people. However, despite the presence of elderly data, the average and standard deviation values show that the global dataset is still made up mostly of young adults, with the percentage of people over 65 years old being just over 20% of the total dataset (37/180 subjects).

Due to the great variability found between datasets, it was necessary to normalize the data from all datasets, according to the normalization procedure depicted in Figure 1a. First, only data corresponding to the acceleration (accelerometer) and angular velocities (gyroscope) of the sensors located in the subjects’ waist were considered. Then, the sensor reorientation method was applied so that the axis orientation corresponded to the one depicted in Figure 1b. Finally, all datasets underwent a resampling process so that the sampling frequency was normalized to 50 Hz.

### 2.2. ADL and Falls

The datasets, whether public or private, contain the great majority of ADL used for activity recognition in the literature. Therefore, a total of 20 labels, including periodic activities, static postures, transitions between postures and falls, were used in order to cover all ADL listed in every dataset. It should be noted that some activities whose labels in public datasets were considered different were recognized as the same activity in this work, since their basic body movement is similar, e.g., the cases of Sisfall’s activity of sitting in high chairs or low chairs were included in the “Stand to Sit” class; or even cases of standing in different places, such as in the room and in the elevator, were all included in the “Standing” class. Table 3 lists the ADL that were adressed in this work.

A study carried out on how the ADL were distributed showed that the global dataset is unbalanced, with a greater tendency toward cyclical activities, such as walking or lying (29.73% and 18.52%, respectively), with only a small percentage of transitions between activities and fall events, such as the fall by syncope, which is the activity with the least amount of data in the constructed dataset (0.27%). Figure 2 shows the percentage amounts of each activity present in the created dataset.

### 2.3. Machine and Deep Learning Classifiers: Comparative Analysis

We performed a comparative analysis, whose strategy is illustrated in Figure 3, to determine the most suitable AI-based classification model and the subset of features that attains the best performance. In the next sections, every step of the proposed strategy will be addressed. In this comparative analysis, the first approach consisted of training several ML classifiers to classify the addressed ADL and fall events with different feature subsets. The subsets of features with the best performance were then used in a second approach, where several Neural Networks architectures were proposed, and their performances were compared with the ML classifiers. Finally, for the best models found between ML and DL approaches, a study to assess the influence of the window size used for feature extraction on the classification model’s performance was carried out. It should be noted that all operations were performed on the global dataset without the use of any noise filters or other sorts of processing, i.e., the raw inertial data were directly implemented in the referred procedures. All the processes used for the development, validation and evaluation of these ADL recognition algorithms were implemented offline using the Matlab 2021b version on a Lenovo Legion Y540: processor—intel core i5, 9th Gen; graphics card—NVIDIA^®^ GeForce^®^ GTX 1650; memory—8 GB DDR4 at 2666 MHz and SSD PCIe of 512 GB.

#### 2.3.1. Feature Extraction

Feature extraction was achieved through the sliding window method, where a signal is segmented into several windows of equal size, on which different features can be calculated. The most used sliding windows’ size corresponded to approximately 1 s for this type of activity classification, and the overlap between consecutive windows can vary from 50% to 87% [1,5,14]. Within the scope of this work, firstly, a one-second window was selected for the comparative analysis, which corresponds to a 50-sample window, with an overlap of 80% (Figure 4). In addition to the initial window size, 4 other different sizes were explored in the window size study: 0.5 s; 1 s; 1.5 s; and 2 s. The overlap was kept at 80% for all tests, despite the literature suggesting that it can also have a high impact on the classification performance and computational cost for real-time applications [1,36]. The segmentation in windows with a size of 1s and an overlap of 80% resulted in a total set of 666,660 windows for the models’ training and evaluation. Similarly, for a window size of 0.5 s, 1.5 s, and 2 s, a total of 1,366,289, 435,111, and 318,516 windows were obtained, respectively.

Thus, for each window, several features, such as the averages, maximums, minimums, and standard deviations of each of the acceleration and angular velocity signals, among other metrics, were extracted, making a total of 199 features calculated. A summary of the extracted features can be seen in Table 4. The window labeling was carried out according to the Mode Labeling Method, where the label of a given window would be the mode of the labels present in that window’s samples [36].

#### 2.3.2. Pre-Processing and Feature Selection

After the aforementioned extraction, min–max scaling [0, 1] was carried out to ensure a low computational cost when building the models [37]. Since each of the datasets collected data with different devices and sensors with various sensitivities, the normalization process was carried out for each public dataset separately in order to reduce the possible bias derived from the different types, ranges, and sensitivities of the sensors used.

In order to understand what kind of improvements this feature selection process can bring, nine Feature Selection Methods (FSM) with diverse types of selection (Filtering, Wrapper and Embedded) were applied to the extracted features [38,39]. Covering the three different types of selection methods mentioned above, the selected methods and their respective type is indicated in Table 5.

Feature selection was also performed by the Principal Component Analysis (PCA), as shown in Figure 5. Principal components (PCs) with a cumulative percent explained of 70% were chosen [47], and a resultant and proportional PC was created and used to rank features. PCA also serves a secondary goal, which is to reduce the computational cost of the comparative analysis. Rather than using all 199 features, the number of features used was decided by multiplying the number of features having a PC value greater than $\frac{1}{199}$ by 2 [34,48]. As a result, we chose a higher number of features for comparative analysis than those chosen for PCA and with significant contributions to the variability of the data. The success of this feature selection procedure will be determined by achieving the best performance with a lower number of features than the number of features extracted (199).

#### 2.3.3. Model Building and Evaluation

We compared the following five ML classifiers by using a set of procedures depicted in Figure 3: DA with linear and quadratic approaches; K-NN with squared inverse distances; Ensemble Learning; and DTs. Table 6 presents a description of these classifiers. A progressive comparative analysis was carried out for each of these classifiers, where using the several FSM (Table 5), it was studied with how many of the ranked features (from most to least important), each of the classifiers obtains the best performance results. As previously mentioned, achieving the best classification performance with a lower number of features than the total number of extracted features in the previous step will show a successful application of this feature selection procedure and progressive analysis.

Initially, we performed the hold-out (HO) method to split 70% of the created dataset’s data for training and 30% for testing. The mentioned classifiers were used to obtain the subset of the most relevant features by performing an initial five-fold cross-validation (CV) with one repetition and using only training data. Once we determined the subset of the most relevant features, we performed a five-fold CV with ten repetitions to the four best classifiers from the previous step in order to evaluate the generalization capabilities of each model. The two best classifiers from this step were chosen, and its hyperparameters were optimized through a gridsearch process. The optimized ML classifiers and the neural networks architectures were further trained with all training data and tested with the test data from the HO method and the final performance of each classifier was compared in order to choose the best AI-based classification models.

In order to understand if these can be a viable alternative to ML methods, the classification of ADL and fall events was also carried out using the four Neural Networks architectures presented in Figure 6. As a result, the two best feature subsets acquired from the validation steps were used as input to train and evaluate the DL models. The performance of neural networks with the two best set of features was analyzed, and from the performance results obtained, a choice between the ML-based and DL-based models was made, for the most suitable classification model for ADL recognition. During all of the operations, test specifications such as the loss function used, number of epochs, the optimizer employed, number of hidden layers, batch size and the Learning Rate were kept constant for all the architectures. Table 7 provides a summary of all these characteristics and respective values.

The last study evaluates which of the chosen window sizes provides better performance results to the selected classifiers. Due to the great diversity in the classes to be classified (static postures, cyclical activities, transitions between postures and falls) and their duration, this study became imperative, in order to find the size that best fits the activities to be classified. Furthermore, the time required to perform the training and testing of the K-NN and Ensemble Learning classifiers for each of the window sizes was also computed. This exercise had the objective of investigating the possibility of using one of the algorithms that presented better performance in real-time situations.

The key performance indicators in the developed work were the Mathew’s correlation coefficient (MCC) and the accuracy (ACC). The first evaluation metric exhibits good representational features of unbalanced classes, as occurs in this study [55]. We used it for model performance comparison and reporting. The ACC, as well as the sensitivity (Sens), specificity (Spec), precision (Prec), and F1-score (F1S), were calculated for benchmarking the findings of the literature.

## 3. Results

### 3.1. PCA Outcomes

We determined that 11 PCs were necessary for a cumulative percent explained greater than 70%. Furthermore, the resulting PC demonstrated that there were 55 features with PC values greater than 1/199. After performing the PCA, we reduced the number of features to lower the computational cost of the comparative analysis (Figure 5); i.e., instead of using all 199 features, we only used the first 110 features ranked by any feature selection method included in this study. Only training data split from HO method were used.

### 3.2. ADL and Fall Events Classification

The results attained from the five-fold CV with one repetition disclosed the Ensemble Learning classifier as the one that presented the best performance among the used classifiers (MCC = 85.78%; ACC = 94.59%) when using the first 65 features ranked by the PCA method (Appendix A, Table A1). With the first 85 features ranked by the Relief-F (Appendix A, Table A1), K-NN produced similar but inferior results (MCC = 85.10%; ACC = 93.63%). DTs performed worse with the first 74 features ordered by the same technique (MCC = 70.65%; ACC = 88.22%); and Quadratic and Linear DA had the worst performance results with the first 55 and 66 features ranked by the Relief-F method, respectively. The two best classifiers went through a five-fold CV with ten repetitions, and we realized that increasing the number of repetitions did not change significantly the CV results either for the Ensemble Learning (MCC = 85.79%; ACC = 94.59%) or K-NN classifier (MCC = 85.05%; ACC = 93.62%). From this second CV stage, the Ensemble Learning using the first 65 features ranked by PCA and the K-NN using the first 85 features ranked by Relief-F were chosen for the next phases. Table 8 presents the main results for the two phases of the CV process.

When using test data from the HO data split method, the two best classifiers presented slight improvements in their performance in comparison to the results shown in Table 8. However, the Ensemble Learning model presented lower results (MCC = 88.36%; ACC = 95.44%) than the K-NN classifier (MCC = 93.19%; ACC = 97.27%) when tested with unseen data, contrary to what was verified during the CV process. After the optimization stage, K-NN hyperparameters were: (i) distance—minkowski; (ii) distance weight—squared inverse; (iii) exponent—0.5; and (iv) number of neighbors—1. Ensemble Learning hyperparameters were: (i) Method—Bag; and (ii) number of learning cycles—37. Table 9 depicts the main results obtained for the HO validation process.

### 3.3. Deep Learning Outcomes

The BiLSTM stood out among the neural networks in both case studies, using the first 85 features ranked by the Relief-F method (MCC = 82.83%; ACC = 92.55%) and the first 65 features ranked by PCA (MCC = 80.52%; ACC = 91.48%). However, in both cases, it presented lower results when compared to Ensemble Learning and K-NN. On the contrary, CNN presented the lowest performance results for both case studies, using the first 85 features ranked by the Relief-F method (MCC = 37.87%; ACC = 57.01%) and the first 65 features ranked by PCA (MCC = 24.90%; ACC = 42.67%). Thus, the ML-based methods, K-NN and Ensemble Learning, were considered the classifiers with better performance for ADL and fall events recognition, among all tested classifiers, being selected for the window size study and classification time analysis. Table 10 contains the main results for the DL-based classification problem.

### 3.4. Window Size and Classification Time

The results attained in this last analysis, for the optimized K-NN and Ensemble Learning classifiers, are depicted in Table 11. It should be noted that the same labeling method was used during the feature extraction step for each window size selected for this analysis. Both tables show a decreasing trend in the performance of the two classifiers as the window size increases from 0.5 to 2 s.

In addition to the results obtained regarding performance metrics, the time required to perform the training and testing of the K-NN and Ensemble Learning classifiers for each of the window sizes used in this last study was also computed. This exercise has the objective of studying the possibility of using one of the combinations which presented better performance results in real-time situations. Table 12 depicts the results obtained for the training and testing time of each classifier and window size combination.

Through direct observation of Table 12, the K-NN classifier has a training time of around four seconds, regardless of the size of the windows. The Ensemble classifier’s training time shows an increasing trend as the window size decreases. On the other hand, the time required to test only one window (last column of Table 12) is lower than $4.5\times 10^{-5}$ s for every window size tested in the case of the Ensemble Learning classification model. The test time per window for the K-NN classifier shows an increasing trend as the window size decreases.

## 4. Discussion

The ML-based classification was established with a combination of different classifiers and feature selection methods, and it proved to be an accurate strategy. This process allowed the most relevant features to be found for the classification of 20 distinct activities and fall classes (Appendix A), which should aid in a time-effective and low computation strategy for real-time ADL recognition. The PCA-based approach for reducing the computational burden of the comparison analysis may therefore be judged effective, since for most of the tested models and feature selection methods, the CV best performance was achieved with fewer features than those initially extracted.

The success achieved regarding the performance of the classification models through different validation methods, namely hold-out and cross-validation, shows the robustness of the applied processes. The creation of a vast dataset also positively influenced the performance results obtained. According to the results presented in Table 11 and Table 12, the Ensemble Learning classifier with a subset of the 65 first features ranked by the feature selection method PCA, with a window of 0.5 s and an overlap of 80%, was the model which showed the most potential for the classification of the 20 ADL classes in real time. Despite not being the best performer in terms of evaluation metrics, it had a classification time lower than the window advance time, allowing, in theory, its deployment in real-time situations. In addition, for the Ensemble Learning classifier, the comparative analysis of the set of features extracted by the different feature selection models provided a reduction in the number of features by more than 65% from the 199 initial features, to a subset of 65 features, also supporting the possibility of applying this type of models in real-time systems.

According to the literature, windows with a larger size have the tendency to perform worst in the recognition of shorter activities or transitions and better in the classification of cyclical or static activities, which are maintained for longer periods of time. Furthermore, as the window size increases, the model’s capability to recognize ADL in real time decreases [1,36]. As stated in Table 11, for smaller windows, the performance of the classifiers analyzed in this dissertation increases, thus corroborating what is described in the literature.

A direct comparison between the two best results obtained in tests performed with Neural Networks (Table 10) and the process developed for ML classifiers (Table 11) shows that the use of different architectures with the same features ranked by the best feature selection methods as inputs did not produce as good results as when using the K-NN and Ensemble Learning classifiers. The tested LSTM and BiLSTM achieved the best results in terms of performance metrics when compared to the CNN and CNN-LSTM architectures as well as in relation to the computation time of training and testing them. As mentioned in the literature, LSTM has a more suitable architecture for classifying sequential data [3,22]. Moreover, as investigated in the literature, the results achieved by BiLSTM were slightly superior to the basic LSTM [5,56]. The 85 features ranked by Relief-f returned performance metrics results slightly higher than the 65 features ranked by the PCA, as what happened in the studies carried out on ML models. CNNs are usually used in problems involving image inputs, and in this process, the implemented CNN architecture received as input 1D arrays of features extracted from sequential data. This method of use may justify the poor results obtained with the CNNs in terms of performance metrics, given that the convolution processes applied to extract features from the arrays used were not carried out in the most effective way. Despite these poor performance results, other architectures involving this type of networks should continue to be tested, with different type of inputs, such as the raw inertial data, since several studies claim to obtain positive results in the recognition of ADL with the use of this type of neural network [8,14,23].

Thus, this initial study of the behavior of several neural networks showed that despite positive results in some of the cases tested, it did not reach the desired potential when compared to the results obtained with the ML classifiers. This fact raises the need to carry out several studies and some changes in the future in order to obtain performance results higher than those found in the literature. The architectures used must be improved in the search for better results in order to be comparable with the studies currently carried out in the literature. Finally, and as mentioned in most of the works analyzed in the literature, new tests should be carried out in the future with windows segmented from raw inertial data as inputs rather than features, given the ability of neural networks to carry out a process extraction of features [23].

## 5. Conclusions and Future Work

An algorithm to recognize sixteen ADL and four different types of fall (twenty classes in total) from several AI-based classification models and feature selection models was built in order to find the combination that presents better performance in this type of classification. The performance of the different combinations was evaluated using the following parameters: (i) performance evaluation metrics, (ii) subset of features used, and (iii) classification time per window of the models.

Two different approaches (ML and DL) were investigated and compared: when performing one waist-located inertial sensor-based ADL and fall events recognition. Furthermore, a new procedure of fusion and normalization of public datasets was carried out in order to generate a big enough dataset to validate the activity classification models in order to battle concerns observed in the literature. Our long-term objective is to evaluate these techniques on elderly waistband users to evaluate whether the results of this study translate to identical outcomes in continuous real-life usage.

Taking into account the performance values as well as the classification times found in this work for the machine used, it is concluded that the most effective AI-based classifier was the Ensemble Learning classifier with the first 65 features ranked by the PCA feature selection method. Moreover, the classification time per window in this combination was lower than the window advance time in every window tested, which represents an encouraging result for the application of this algorithm in real time in the future. The DL outcomes were not as good as in the prior procedure; however, their potential was demonstrated, indicating that they could be a good option in the future with the appropriate future work regarding the input data used and its architectures. Although the established methodology allows for a reduction in computing cost, we recommend using a sufficient processor so that all classifiers respond rapidly.

The constructed dataset contains a large amount of successfully normalized data for the validation of activity recognition algorithms. Different steps can still be applied for its continuous improvement, such as the addition of more senior subjects’ data and data-balancing techniques, such as data augmentation, to balance the amount of samples of ADL represented on a smaller scale. However, the choice between using balanced data or preserving the overall distribution of activities should be backed by a critical analysis of the results obtained through a validation with data collected in day-to-day circumstances over long periods of time. Data fusion from other sensors, such as magnetometers or barometers, as well as the addition of other devices for data collection (such as smartwatches, which do not impose any restrictions on the subject’s movements), may aid in the differentiation of activities with similar movement patterns. Improved data-splitting methods must be verified in the future to guarantee that the results attained were not due to the usage of comparable data during the train and test stages of the model. Despite attaining performance results comparable with the literature for bidirectional LSTM, considering the higher number of classes classified, continued development of the tested neural networks is needed. New and improved architectures, as well as ablation studies, should also be conducted to determine the impact of the various settings and stages of each architecture on ADL and fall events recognition. Furthermore, the constructed classifiers should be trained and tested on the various public datasets independently in order to make a more direct comparison between the methods covered in this study and those listed in the literature that use the same datasets for evaluation.

This study contributes to the state-of-the-art with two versatile ADL and fall events detection methods, which are capable of discriminating 20 classes of events. There is also evidence that dataset fusion and normalization is imperative to guarantee a vast and diverse amount of data for ADL recognition algorithms’ validation. Furthermore, it may lead to the incorporation of these tools on instrumented waistbands or other devices for real-time fall risk assessment in the future [57].

## Figures and Tables

**Figure 1 sensors-22-04028-f001:**
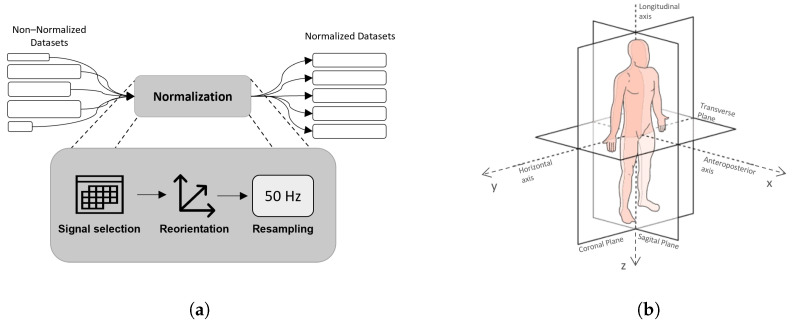
(**a**) Normalization process steps implemented in order to normalize the public datasets for ADL recognition. (**b**) Desired sensors orientation. The inertial sensors have to be located on the subjects’ waist. The arrows and letters x, y and z indicate the positive direction of the anteroposterior, mediolateral and longitudinal axes, respectively.

**Figure 2 sensors-22-04028-f002:**
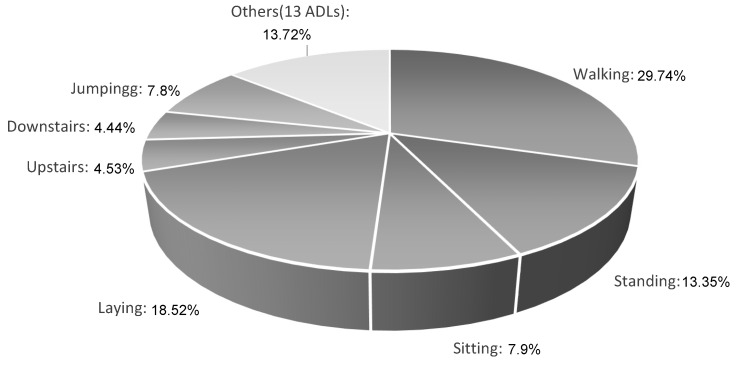
Percentage quantity of windows created of each activity present in the created dataset. The activities are named according to Table 3.

**Figure 3 sensors-22-04028-f003:**
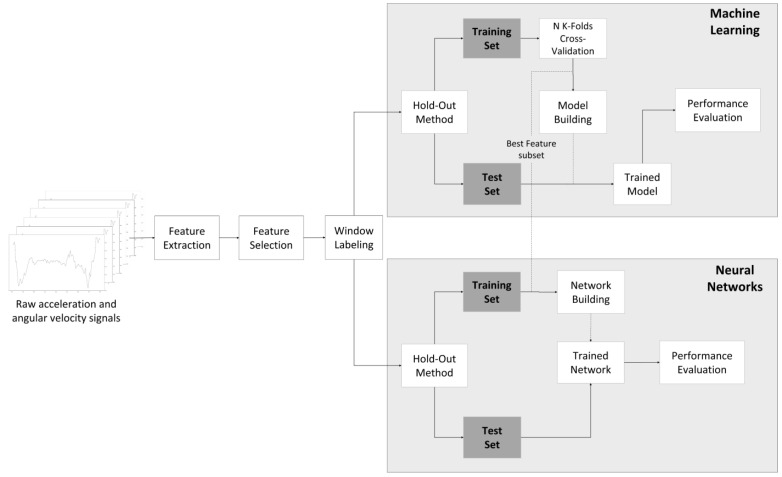
Complete process for all the validation, training and evaluation of different Machine Learning and Deep Learning models alongside the best feature set selected by diverse feature selection methods.

**Figure 4 sensors-22-04028-f004:**
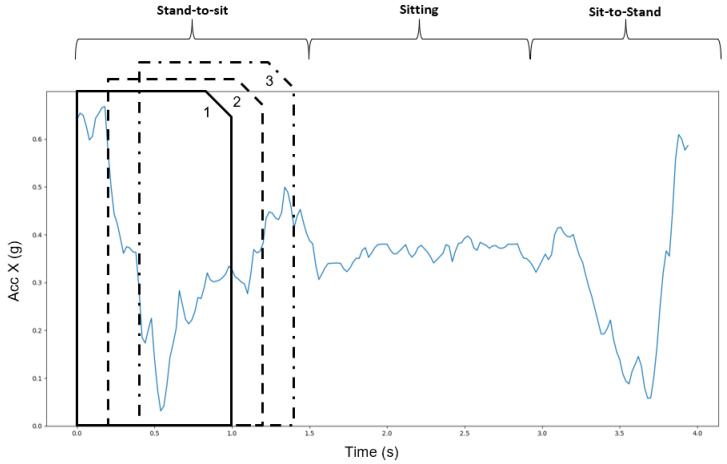
Signal segmentation example, using the sliding window technique example for feature extraction. A sliding window across time is represented by windows 1 to 3.

**Figure 5 sensors-22-04028-f005:**
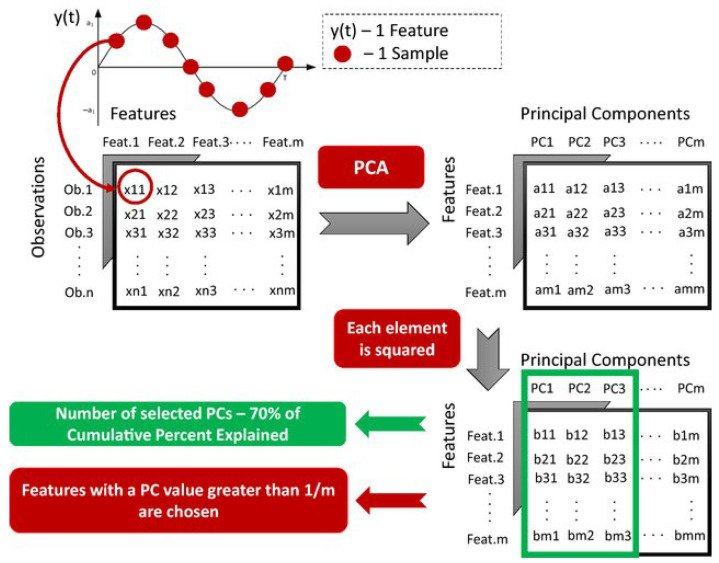
PCA-based procedure to rank and obtain the most crucial features and to limit the computational cost of comparative analysis.

**Figure 6 sensors-22-04028-f006:**
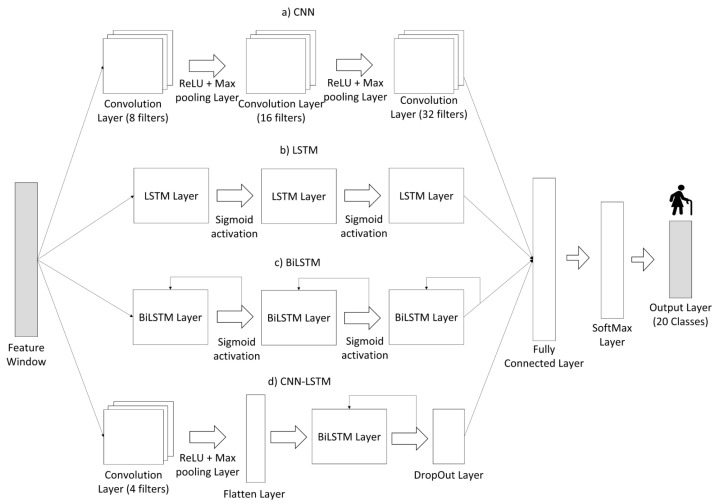
Neural Networks architectures used in this manuscript. The input shown represents a single feature window. (**a**) The CNN identifies correlations from the various features provided. (**b**) The LSTM network detects crucial temporal features. (**c**) The BiLSTM network has a similar operating mode as LSTM but with bidirectional LSTM layers. (**d**) The hybrid CNN-LSTM extracts temporal patterns using convolutional features from the CNN convolutional layer. Based on [53].

**Table 1 sensors-22-04028-t001:** Datasets used for the evaluation of the Deep Learning algorithms analyzed in the literature and respective description regarding sensing methods, sample frequency, participants, number of activities (classes) recorded and algorithm performance (accuracy). In this table: A = Accelerometer, G = Gyroscope, M = Magnetometer, B = Barometer and ADL = Activities of Daily Living.

DataSet	Work	Sensors	Sample Frequency	Participants	N° of Classes	Accuracy
Private Dataset	Chung et al. [22]	A, G, M	100 Hz	5	9	93%
SisFall [16]	Wang et al. [8]	A, G	200 Hz	38	2	<99%
PAMAP2 [18]	Gil-Martín et al. [23]	A, G, M	9 Hz	9	18	96.62%
UCI-HAD [15]	Altuve et al. [2] Murad et al. [3]	A, G, M	50 Hz	30	6	[2]-96.7% [3]-92.9%
USC-HAD [24]	Murad et al. [3]	A, G	100 Hz	14	12	97.8%
Opportunity [17]	Murad et al. [3]	A, G, M	30 Hz	4	18 ^1^	92.5%
Daphnet FOG [25]	Murad et al. [3]	A	64 Hz	10	2	94.1%
Skoda [26]	Murad et al. [3]	A	98 Hz	1	11 ^1^	92.6%

^1^ ADL related to hand gestures.

**Table 2 sensors-22-04028-t002:** Datasets description regarding sensing methods and location, sample frequency, participants and activities recorded, where: A = Accelerometer, G = Gyroscope, M = Magnetometer, B = Barometer and ADL = Activities of Daily Living.

DataSet	Availability	Sensors	Location	Sample Frequency	Participants	ADL Falls
SisFall [16] ^1^	Public	A, G	Waist	200 Hz	23 subjects <30 years 15 subjects >60 years	19 ADL 15 Falls
FALLALLD [29] ^1^	Public	A, G M, B	Chest, Waist, Wrist	238 Hz	15 subjects 21–53 years	44 ADL 35 falls
FARSEEING [30] ^1^	Public	A, G, M	Waist, Thigh	20 Hz 100 Hz	20 subjects ^2^	Real falls
UCI HAR [15]	Public	A, G	Waist	50 Hz	30 subjects 19–48 years	12 ADL
Cotechini [31] ^1^	Public	A, G	Waist	33, 33 Hz	8 subjects	5 ADL 13 Falls
UMAFall [32]	Public	A, G, M	Waist, Chest, Wrists, Ankle, Front pocket	20 Hz	17 subjects 18–55 years	8 ADL 3 Falls
+Sense [33]	Private	A, G, M	Waist	100 Hz	10 Healthy 40 Pathological	1 ADL
SafeWalk [34]	Private	A, G, M	Waist, Thighs, Feet	30 Hz	12 subjects 25.33 ± 6.33 years	1 ADL Fall
InertialLab [35]	Private	A, G, M	Waist, Thighs, Shank, Feet	200 Hz	7 subjects 23–26 years	5 ADL

^1^ Several activities in these datasets were grouped into one single class of basic activities. ^2^ Only data from 3 subjects were suitable to use.

**Table 3 sensors-22-04028-t003:** Static postures and locomotion daily activities, postural transitions and fall events selected to be recognized by the Machine and Deep Learning models.

Periodic Activities and Static Postures	Transitions	Fall Events
Walking	Lying to Stand	Forwards
Standing	Stand to Sit	Backwards
Sitting	Sit to Stand	Lateral
Lying	Stand to Pick to Stand	Syncope
Upstairs	Stand to Lying	
Downstairs	Change Position (Lying)	
Jumping	Turning	
Jogging	Bending	

**Table 4 sensors-22-04028-t004:** List of all extracted features from each window created. AP, V and ML refer to the anteroposterior, vertical and mediolateral axis, respectively.

Feature Number	Feature Description
[1–6]	Acceleration and Angular velocity (AP, V, ML)
[7–8]	SumVM of acceleration and Angular velocity
[9–24]	Skewness and kurtosis of acceleration, Angular velocity (AP, V, ML) and SumVM signals
[25–64]	Min, max, mean, variance and Std deviation of acceleration, angular velocity (AP, V, ML) and SumVM signals
[65–70]	Correlation between V-ML, V-AP and ML-AP axis of acceleration and Angular velocity
[71–77]	Slope, Total angular change, Resultant angular acceleration, ASMA, SMA, Absolute vertical acceleration, Cumulative horizontal displacement
[78–102]	Peak-to-Peak, Root Mean Square and Ratio Index of Acceleration, Angular velocity (AP, V, ML) and SumVM signals
[103–115]	Resultant angle change, Flutuation frequency, Resultant of average acceleration and Resultant of standard deviation (AP, V, ML)
[116–117]	Resultant of Delta changes of acceleration and Angular velocity
[118–133]	Gravity component, Displacement, Displacement range, Cumulative sway length and Mean sway velocity (AP, V, ML)
	Slope changes, Zero crossings, Waveform length of acceleration, Angular velocity (AP, V, ML) and SumVM signals
[133–189]	Energy, Mean frequency, Peak frequency and magnitude of acceleration, Angular velocity (AP, V, ML) and SumVM signals
[190–195]	SumVM of resultant angular velocity, average acceleration and Standard deviation, Maximum resultant angular velocity and Acceleration in the horizontal plane
[196–199]	Acceleration exponential moving average, Rotational angle of acceleration SumVM, Z-Score, Magnitude of angular displacement

**Table 5 sensors-22-04028-t005:** Feature Selection Methods Tested for ADL Recognition.

Feature Selection Methods (FSM)	FSM Type
Infnite Latent Feature Selection (ILFS) [38]	Filtering
Unsupervised Feature Selection with Ordinal Locality (UFSOL) [40]	Wrapper
Feature Selection with Adaptive Structure Learning (FSASL) [41]	Wrapper
Minimum-Redundancy Maximum-Relevancy (MRMR) [42]	Filtering
Relief-F [43]	Filtering
Mutual Information Feature Selection (MutInfFS) [44]	Filtering
Feature Selection Via Concave Minimization (FSV) [45]	Embedded
Correlation-Based Feature Selection (CFS) [46]	Filtering
Least Absolute Shrinkage and Selection Operator (LASSO) [43]	Embedded
Principal Component Analysis (PCA) [47]	Filtering

**Table 6 sensors-22-04028-t006:** Machine Learning models evaluated in this work and respective descriptions.

Model	Reference	Description
DA	[49]	A method that finds combinations of features that separate two or more classes of objects or events, searching for the most variance between classes, and information that maximizes the difference between classes.
K-NN	[50]	Compares each new instance with all datasets available and the instance closest by distance metrics is used to perform classification. Since every sample of the dataset must be checked for every instance, the time and complexity of the method rises according to the dataset size.
Ensemble Learning	[51]	Creates multiple instances of traditional ML methods and combines them to evolve a single optimal solution to a problem. This approach is capable of producing better predictive models compared to the traditional approach.
DTs	[52]	A model that predicts the value of a target variable based on numerous input variables. A decision tree is constituted by an internal node, based on which the tree splits into branches. The end of the branch that does not split any longer is the decision.

**Table 7 sensors-22-04028-t007:** Specifications for the use of the Deep Learning models depicted in Figure 6.

Specification	Value
Epoch Number	100
Hidden Layers	150
Batch Size	64
Optimizer	Adam [54]
Learning Rate	0.001 (Constant)
Loss Function	Cross-Entropy

**Table 8 sensors-22-04028-t008:** Comparison of the best classification results (ACC, Sens, Spec, Prec, F1S, MCC), attained after the 5-1 and 5-10 k-fold cross-validation steps for the K-NN and Ensemble Learning classifiers.

ML Model	FSM	CV Step	N° of Features	ACC (%)	Sens (%)	Spec (%)	Prec (%)	F1S (%)	MCC (%)
K-NN	Relief-F		85	93.63	84.17	99.64	86.80	85.43	85.10
	PCA	5 Fold (1 rep.)	85	92.99	84.08	99.60	86.01	85.01	84.63
	FSASL		70	91.49	81.39	99.51	83.66	82.48	82.02
Ensemble Learning	PCA		65	94.59	82.22	99.68	90.54	85.80	85.78
K-NN	Relief-F		85	93.62 ± 0.016	84.12 ± 0.066	99.64 ± 0.001	86.75 ± 0.055	85.38 ± 0.056	85.05 ± 0.056
	PCA	5 Fold (10 rep.)	85	92.95 ± 0.021	83.91 ± 0.094	99.60 ± 0.001	85.88 ± 0.085	84.86 ± 0.085	84.48 ± 0.086
	FSASL		70	91.48 ± 0.026	81.40 ± 0.063	99.51 ± 0.001	83.59 ± 0.079	82.45 ± 0.066	81.99 ± 0.067
Ensemble Learning	PCA		65	94.59 ± 0.015	82.18 ± 0.067	99.68 ± 0.001	90.64 ± 0.073	85.79 ± 0.061	85.79 ± 0.060

**Table 9 sensors-22-04028-t009:** Hold-out test results for the Ensemble Learning with the first 65 features ranked by the PCA and for the K-NN classifier with the first 85 features ranked by the Relief-f.

ML Model	FSM	N° of Features	ACC (%)	Sens (%)	Spec (%)	Prec (%)	F1S (%)	MCC (%)
K-NN	Relief-F	85	97.27	92.90	99.84	93.79	93.34	93.19
Ensemble Learning	PCA	65	95.44	85.97	99.73	91.67	88.43	88.36

**Table 10 sensors-22-04028-t010:** Results for the test of the 4 Deep Learning architectures with the 85 first features ranked by Relief-f and 65 first features ranked by PCA.

FSM	Feature Number	Architecture	ACC (%)	Sens (%)	Spec (%)	Prec (%)	F1S (%)	MCC (%)
Relief-F	85	CNN	57.01	37.06	97.22	54.67	35.47	37.87
		LSTM	92.06	79.58	99.55	84.25	81.02	81.01
		CNN-LSTM	88.84	74.48	99.36	75.24	74.53	74.06
		BiLSTM	92.55	81.14	99.57	85.56	83.14	82.83
PCA	65	CNN	42.67	26.46	96.15	54.49	22.27	24.90
		LSTM	91.46	77.81	99.51	84.38	80.61	80.38
		CNN-LSTM	88.55	74.33	99.35	75.09	74.36	73.88
		BiLSTM	91.48	79.33	99.52	83.32	80.67	80.52

**Table 11 sensors-22-04028-t011:** Window size comparative study results for the K-NN best optimized model with the Relief-F feature selection model.

ML Model + FSM	Window Size (s)	Window Overlap (%)	ACC (%)	Sens (%)	Spec (%)	Precn (%)	F1S (%)	MCC (%)
K-NN + Relief-f	0.5	80	98.22	95.20	99.90	96.04	95.62	95.52
	1		97.27	92.90	99.84	93.79	93.34	93.19
	1.5		96.30	91.73	99.79	91.15	91.41	91.22
	2		95.33	90.53	99.74	88.51	89.44	89.22
Ensemble + PCA	0.5		96.53	88.94	99.79	94.09	91.29	91.21
	1		95.44	85.97	99.73	91.67	88.43	88.36
	1.5		95.01	85.60	99.71	90.76	87.64	87.62
	2		94.51	85.21	99.68	89.37	86.92	86.79

**Table 12 sensors-22-04028-t012:** Classification time for the training and testing of the two best combinations of Machine Learning (ML) model and Feature Selection Method (FSM), for each of the selected windows for the window size study.

ML Model + FSM	Window Size (s)	Window Overlap (%)	Test Windows	Train Time (s)	Test Time (s)	Test Time per Window (s)
K-NN + Relief-f	0.5	80	409,740	4.36	213,588.88	0.521
	1		199,997	4.18	66,782.58	0.334
	1.5		130,421	4.70	12,633.08	0.097
	2		95,482	4.09	6752.47	0.071
Ensemble + PCA	0.5		409,740	829.55	15.99	$3.90\times 10^{-5}$
	1		199,997	279.03	8.54	$4.27\times 10^{-5}$
	1.5		130,421	145.21	5.68	$4.35\times 10^{-5}$
	2		95,482	100.23	3.94	$4.13\times 10^{-5}$

## Data Availability

Not applicable.

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
