# Peer review of "Inertial Data-Based AI Approaches for ADL and Fall Recognition"

_sensors, 2022, doi:10.3390/s22114028_

Round 1
Reviewer 1 Report
This paper uses the machine learning and deep learning technology to recognize the Activities of Daily Living (ADL), especially fall. As such, the matter is of interest. However the paper suffers for several serious limits:
- What abbreviations are line 15, PCA, line 19, LSTM, etc? Need to write the full name.
- Machine learning is a data-driven method. The amount of data in this manuscript is not enough, so more data need to be extracted for processing.
- Some databases contain some non-health groups (such as Parkinson's patients). How is this data processed and integrated? Please give more details?
- The manuscript introduces a large number of conventional image processing methods, similar to a technical manual, which is not suitable for publication as a paper.
- Bi-lstm algorithm is one of the commonly used algorithms. This manuscript does not improve the algorithm itself, the recognition accuracy is not very high, and its innovation is not enough.
In short, this manuscript analyzes and integrates the relevant data sets and creates its own data set, but the amount of data in this data set is too small, the image processing method and deep neural network algorithm used have not been improved, and its innovation is not enough.
Author Response
Reply to Reviewers’ report:
- Reviewer #1:
This paper uses the machine learning and deep learning technology to recognize the Activities of Daily Living (ADL), especially fall. As such, the matter is of interest. However the paper suffers for several serious limits:
Comment 1.1: __________________________________________________________
What abbreviations are line 15, PCA, line 19, LSTM, etc? Need to write the full name.
Answer to comment 1.1:
Thank you for this comment. Effectively, it was a mistake to use only the abbreviations, the suggestion was followed, and the full names of the abbreviations were added, namely:
« However, Ensemble Learning classifier with the first 65 features ranked by Principal Component Analysis (PCA)… (overall accuracy of 92.55\% for Bidirectional Long Short-Term (LSTM) Neural Network) »
Comment 1.2: __________________________________________________________
Machine learning is a data-driven method. The amount of data in this manuscript is not enough, so more data need to be extracted for processing.
Answer to comment 1.2:
The authors appreciate this comment, which raises a general problem in the analyzed literature, the lack of data for the creation of robust machine learning models. Our strategy to tackle this problem was to merge and normalize 9 datasets describing ADLs (6 public and 3 private) as mentioned in section 2.1. In total, a global dataset was collected containing data from 180 subjects, an amount that was not found in no other work in the analyzed literature for Machine and Deep learning studies.
As stated in manuscript’s section "1. Introduction", the largest public dataset used in the analyzed studies found was SisFall (reference Sucerquia et. al. [16]), with 38 subjects. For example, Murad et. al. (2017) [3] evaluated their algorithms in several public datasets, where the maximum number of windows used for model’s training and evaluation was 57012 and 14253, respectively. In this manuscript, for a window with a size of 1s and 80% overlap, with a sampling rate of 50 Hz, a total number of 666660 windows were created for the algorithm’s training and testing. A table (Table 1) describing the datasets used in the analyzed literature was added to this manuscript for clarity.
However, we understand your comment. Therefore, a better description of the amount of data used to train and test the models was added in section “2.3.1. Feature extraction”, as follows:
«The segmentation in windows with a size of 1s and an overlap of 80% resulted in a total set of 666660 windows for the models’ training and evaluation. Similarly, for a window size of 0.5s, 1.5s and 2s, a total of 1366289, 435111 and 318516 windows were obtained, respectively. »
Also, recommendations regarding the amount of data were referred to in section “5. Conclusions and Future Work”, as can be seen below:
«The constructed dataset contains a large amount of successfully normalized data for the validation of activity recognition algorithms. Different steps can still be applied for its continuous improvement, such as the addition of more senior subjects’ data and data balancing techniques… »
Comment 1.3: __________________________________________________________
Some databases contain some non-health groups (such as Parkinson's patients). How is this data processed and integrated? Please give more details?
Answer to comment 1.3:
Thank you again for this comment. It allowed us to understand the need to improve the manuscritp's message regarding the process of choosing and merging datasets. The addition of data from unhealthy subjects was carried out in order to increase the diversity and size of the dataset, both in terms of data variability (different types of gait) and in terms of age (the aforementioned dataset contains data from elderly people), while maintaining high performances. All data, regardless of whether were data from healthy or unhealthy subjects, were processed in the same way, in order to normalize the data from the different datasets.
Therefore, in order to improve the clarity of the document, these explanations were added in section “2.1. Public Dataset Fusion and Normalization”, as can be seen below:
«The fusion of these various datasets, which include information from both adult and elderly subjects, as well as healthy and unhealthy individuals, served two purposes. First, to create a large and diverse dataset suitable for training ML models. The second goal was to develop AI-based models capable of recognizing ADLs regardless the subject’s age and health condition. »
Comment 1.4: __________________________________________________________
The manuscript introduces a large number of conventional image processing methods, similar to a technical manual, which is not suitable for publication as a paper.
Answer to comment 1.4:
The authors appreciate this comment, which made us recognize that the manuscript needs to be further clarified. The manuscript attempts to classify 20 ADLs using inertial data from participants' waists; and uses some image processing algorithms, such as CNNs.
These were selected in this study because they have been utilized successfully in previous studies, either for fall detection (reference Wang et al. [8]) or for ADL recognition (reference Gil-Martn et al. [23]).
The use of this type of neural network was mentioned in section “1. Introduction”:
«Experiment results attained by Wang et al. [8] show that lightweight neural networks have obtained better results than ML methods previously used in fall detection. The best result was attained with the supervised Convolutional Neural Network (CNN)… A 3-part modular system based on CNNs was proposed by Gil-Martín et al. [23] in order to classify 12 different physical activities... »
Comment 1.5: __________________________________________________________
Bi-lstm algorithm is one of the commonly used algorithms. This manuscript does not improve the algorithm itself, the recognition accuracy is not very high, and its innovation is not enough.
Answer to comment 1.5:
Thank you for this comment. It allowed us to improve the manuscript's message regarding its proposed goals and algorithms. The purpose of using deep learning methods was to make a comparison with the more classic machine learning methods used initially, and that is why simpler neural network architectures were used, taking into account the type of networks most found in the literature. Despite this simpler use, our Bi-LSTM architecture still achieved a classification accuracy higher than 92% for 20 different ADLs, a number superior to those found in the literature.
However, we appreciate the comment, and in order to further clarify why the use of common deep learning algorithms, some comments were discussed and added to the manuscript in section “5. Conclusions and Future Work”, as follows:
«Despite attaining performance results comparable with the literature for Bidirectional LSTM, considering the higher number of classes classified, continued development of the tested neural networks is needed. New and improved architectures, as well as ablation studies, should also be conducted to determine the impact of the various settings and stages of each architecture on ADL and fall events recognition. »
Reviewer 2 Report
The paper is very well written, and a reader can easily follow it. The motivation behind the paper is clear, and so are the contributions. The proposed framework is presented with enough information. The experiment is conducted correctly; the results are well described. Used references are relevant and up-to-date.
There are just a few minor things that I suggest adding.
Is the newly created dataset somewhere published? If not, I suggest doing so.
The authors must provide more explanation about the used parameters. Did they test different settings?
Statistical analysis at the end would be very beneficial for the paper.
Author Response
- Reviewer #2:
Comments: ____________________________________________________________
The paper is very well written, and a reader can easily follow it. The motivation behind the paper is clear, and so are the contributions. The proposed framework is presented with enough information. The experiment is conducted correctly; the results are well described. Used references are relevant and up-to-date. There are just a few minor things that I suggest adding.
Comment 2.1: __________________________________________________________
Is the newly created dataset somewhere published? If not, I suggest doing so.
Answer to comment 2.1:
We greatly appreciate this suggestion. The dataset used in this article was built by fusing several datasets that are already publicly available, as well as other team-own datasets that are private, but associated with publications mentioned in the manuscript. At the moment, it will not be possible to make these datasets available. However, the authors undertake to make them public as soon as possible.
Comment 2.2: __________________________________________________________
The authors must provide more explanation about the used parameters. Did they test different settings?
Answer to comment 2.2:
The authors appreciate this feedback since it should be clear which parameters and settings are used. Deep learning architectures based on literature (reference 51) were used in this paper to compare them to Machine Learning models and validate their potential.
Thus, the parameters and settings shown in figure 6 and table 6 were used in this paper, and they were kept constant throughout all testing.
In any case, we acknowledge that in order to get better performance results in the future, improvements in architectures as well as the settings employed are required, as described in section “5. Conclusions and Future Work”:
«…continued development of the tested neural networks is needed. New and improved architectures, as well as ablation studies, should also be conducted to determine the impact of the various settings and stages of each architecture on ADL and fall events recognition. »
Comment 2.3: __________________________________________________________
Statistical analysis at the end would be very beneficial for the paper.
Answer to comment 2.3:
Thank you for this comment. More statistical information regarding the age and gender of the subjects that make up the dataset were added in section “2.1. Public Dataset Fusion and Normalization”:
«… Furthermore, the global dataset presents a balanced distribution regarding the subject's gender (M = 54%, F = 46%), also containing data from both young adults and elderly people. However, despite the presence of elderly data, the average and standard deviation values show that the global dataset is still made up mostly of young adults, with the percentage of people over 65 years old being just over 20% of the total dataset (37/180 subjects). »
Furthermore, other statistical data such as the standard deviations of the 5-fold cross-validation process with 10 repetitions were added to table 7.
Reviewer 3 Report
The manuscript presents an evaluation study of different Machine Learning and Deep Learning models to classify ADLs and falls, using the fusion and normalization of different datasets present in the literature.
The manuscript is well written and technically focused; however, some minor points should be addressed.
1) Consider shortly describing the validated machine learning and deep learning methods.
2) In Figure 3, for the validation of different Machine Learning and Deep Learning models, the authors considered the same process based on the definition of handcrafted features. Why did the authors not consider the feature learning ability of deep neural networks?
3) What activation functions were used for the LSTM and BiLSTM networks represented in Figure 6 (cases b) and c))?
4) The authors merged and normalized several datasets to validate various Machine Learning and Deep Learning models. However, it would be interesting to see a comparison with similar studies in the literature based on single datasets.
SPECIFIC COMMENTS
Line 106
"Stan-to-Sit" should read "Stand-to-Sit"
Lines 106-107
"Stand-to-Lyin" should read "Stand-to-Lying"
Line 115 and Line 338
Quote: "KNN"
Comment: For the sake of consistency, please use the same acronym k-NN as previously defined.
Table 1
"ADL = Activities of th Daily Living" should read "ADL = Activities of Daily Living"
Line 340
"Table 6" should read "Table 7"
Author Response
- Reviewer #3:
Comments: ____________________________________________________________
The manuscript presents an evaluation study of different Machine Learning and Deep Learning models to classify ADLs and falls, using the fusion and normalization of different datasets present in the literature.
The manuscript is well written and technically focused; however, some minor points should be addressed.
Comment 3.1: __________________________________________________________
Consider shortly describing the validated machine learning and deep learning methods.
Answer to comment 3.1:
Thank you for this comment, which we believe that helped in better organising the methods section and present a better description of the methods used. A new table (Table 5) with short descriptions of the Machine Learning methods was added to section “2.3.3 Model building and evaluation” in order to better explain to the readers the used methods. Figure 6 shows the 4 deep learning architectures used in this paper.
Comment 3.2: __________________________________________________________
In Figure 3, for the validation of different Machine Learning and Deep Learning models, the authors considered the same process based on the definition of handcrafted features. Why did the authors not consider the feature learning ability of deep neural networks?
Answer to comment 3.2:
We appreciate your comment. We effectively recognize the ability of deep neural networks to learn features without having to be handcrafted. The option to use manually determined features was with the intention of comparing results between machine learning models and deep learning models and studying the behavior of neuronal networks with the features considered most important by feature selection methods.
Despite this, our Bi-LSTM architecture still achieved a classification accuracy higher than 92% for 20 different ADLs, a number superior to those found in the literature. This performance was not significantly worse than the used Machine Learning approach. This means that in the future, with a different type of application of these architectures, even higher performance might be attained.
However, there is indeed a need for improvements in the use of deep learning algorithms, which were discussed and added to the manuscript in section “5. Conclusions and Future Work”, as follows:
«Despite attaining performance results comparable with the literature for Bidirectional LSTM, considering the higher number of classes classified, continued development of the tested neural networks is needed. New and improved architectures, as well as ablation studies, should also be conducted to determine the impact of the various settings and stages of each architecture on ADL and fall events recognition. »
Comment 3.3: __________________________________________________________
What activation functions were used for the LSTM and BiLSTM networks represented in Figure 6 (cases b) and c))?
Answer to comment 3.3:
The authors agree that it should be made more clear which activation functions are used. Standard Sigmoid functions were used for both LSTM and BiLSTM. These functions were indicated in a new Figure 6 of the manuscript, whose legend is highlighted in orange.
Comment 3.4: __________________________________________________________
The authors merged and normalized several datasets to validate various Machine Learning and Deep Learning models. However, it would be interesting to see a comparison with similar studies in the literature based on single datasets.
Answer to comment 3.4:
Thank you for your input. We agree that, in terms of benchmarking, it would be an excellent approach to directly compare the used algorithms with others published in the literature. The goal of this study was to counteract the lack of data for the design of strong algorithms by demonstrating to the scientific community a strategy to build large datasets and work around this difficulty. Because of the changes made in each dataset in terms of picking the intended daily activities and labeling these same activities in a normalized manner, comparing datasets is challenging at this moment, and it is necessary to reformulate the processing conducted on the data for such comparison.
This issue was addressed under the section “5 Conclusions and Future Work” with the following:
«Furthermore, the constructed classifiers should be trained and tested on the various public datasets independently, in order to make a more direct comparison between the methods covered in this study and those listed in the literature that use the same datasets for evaluation. »
SPECIFIC COMMENTS: ________________________________________________
Thank you for these comments, which we believe helped to improve the consistency of the nomenclature used throughout the paper. All the mentioned typos were corrected and highlighted in the manuscript.
Round 2
Reviewer 1 Report
Although the manuscript has not made substantive improvements to comment 5 and the innovation of the algorithm is slightly insufficient, the manuscript has made a detailed discussion on the development of relevant fields, and its application effect is also acceptable. In view of this, I agree to accept the manuscript.